# Challenges of Storage and Stability of mRNA-Based COVID-19 Vaccines

**DOI:** 10.3390/vaccines9091033

**Published:** 2021-09-17

**Authors:** Mohammad N. Uddin, Monzurul A. Roni

**Affiliations:** 1College of Pharmacy, Mercer University, Atlanta, GA 30341, USA; uddin_mn@mercer.edu; 2College of Medicine, University of Illinois, Peoria, IL 61605, USA

**Keywords:** COVID-19, mRNA vaccine, SARS-CoV-2, stability, storage

## Abstract

In December 2019, a new and highly pathogenic coronavirus emerged—coronavirus disease 2019 (COVID-19), a disease caused by severe acute respiratory syndrome coronavirus-2 (SARS-CoV-2), quickly spread throughout the world. In response to this global pandemic, a few vaccines were allowed for emergency use, beginning in November 2020, of which the mRNA-based vaccines by Moderna (Moderna, Cambridge, MA, USA) and BioNTech (BioTech, Mainz, Germany)/Pfizer (Pfizer, New York, NY, USA) have been identified as the most effective ones. The mRNA platform allowed rapid development of vaccines, but their global use is limited by ultracold storage requirements. Most resource-poor countries do not have cold chain storage to execute mass vaccination. Therefore, determining strategies to increase stability of mRNA-based vaccines in relatively higher temperatures can be a game changer to address the current global pandemic and upcoming new waves. In this review, we summarized the current research strategies to enhance stability of the RNA vaccine delivery system.

## 1. Introduction

Coronavirus disease 2019 (COVID-19) was first identified in December 2019 in Wuhan, China. Within 18 months, it became the greatest pandemic in modern time. As of September 2021, more than 4.5 million people have died from COVID-19, and more than 215 million people have been infected by it [1]. From November 2020 to August 2021, at least six vaccines have been introduced globally [2,3,4]. Among the COVID-19 vaccines distributed in the United States, messenger RNA (mRNA)-based vaccines developed by Moderna and BioNTech/Pfizer were found to have the highest efficacy rate [5,6]. The mRNA vaccines are preferred, particularly in a pandemic, over traditional vaccines, as they can be rapidly developed, with faster manufacturing times [7]. The mRNA vaccines encode for SARS-CoV-2 antigen (spike protein) using the host ribosomes. The viral spike protein is an ideal target for vaccines due to its critical role in attachment to mammalian ACE2 receptors [3]. The spike protein induces T cell response in the body [8]. Unlike DNA COVID-19 vaccines from AstraZeneca (AstraZeneca, Cambridge, UK) or Johnson & Johnson (Johnson & Johnson, New Brunswick, NJ, USA), mRNA vaccines do not deliver the viral nucleic acid to the host nucleus and, thus, avoid integration with the host DNA [9,10]. The injected mRNA also has a short half-life in vivo and it is removed from the body in a few days [11]. Different COVID-19 variants can also be covered by one mRNA vaccine. Clinical trials are currently underway for mRNA vaccines which could provide protection against emerging variants of SARS-CoV-2 (ClinicalTrials.gov (accessed 15 September 2021) Identifier: NCT04785144).

Despite these merits of efficacy and safety, instability and ultracold storage requirement of mRNA vaccines remain major limitations. The stability of this emerging and fast-growing vaccine platform is poorly understood, and it likely depends on multiple factors, such as excipients, pH, and temperature [12].

Stability of mRNA vaccines can be impacted, to some extent, by encapsulating mRNA in lipid nanoparticles (LNP) [13,14]. Although vaccines from different manufacturers use LNP as a carrier for mRNA, their storage conditions are different (Table 1). BioNTech/Pfizer COVID-19 vaccine requires storage at −80 °C with a shelf life up to 6 months, whereas the Moderna COVID-19 vaccine requires storage at −20 °C with the same shelf life. Special packaging with dry ice is used for the BioNTech/Pfizer vaccine during transport. The difference in storage conditions is likely related to extra precautions by BioNTech/Pfizer. Recent stability study data submitted by Pfizer to the European Medicines Agency (EMA) suggests that BioNTech/Pfizer vaccines have similar shelf life (30 days) to Moderna’s vaccine when refrigerated at 2–8 °C [15]. Apart from this recently released storage guideline from the EMA, further stability data are not available in the public domain.

Another LNP-based mRNA vaccine CVnCoV was developed by CureVac (CureVac, Tübingen, Germany) which can be stored at much warmer temperatures (5 °C) for at least three months (Table 1) [13], indicating better thermostability compared to vaccines from Moderna or BioNTech/Pfizer. CureVac’s vaccine used the same LNP platform of Pfizer/BioNTech with a different kind of synthetic mRNA. The mRNA (encoding viral spike protein) used by CureVac’s COVID-19 vaccine did not go through nucleoside modification like its rivals [16]. Unfortunately, CureVac’s COVID-19 vaccine candidate, which used a smaller dose (12 µg) than Pfizer’s (30 µg), showed poor efficacy (47%) in the phase 2b/3 trial [17]. The reasons for the unexpected low efficacy are unclear. A new mRNA LNP vaccine developed by Walvax (Walvax, Kunming, China) (ARCoV), undergoing a phase 3 clinical trial (ClinicalTrials.gov Identifier: NCT04847102), showed promising thermostability. This vaccine employed nucleoside modified the mRNA encoding viral receptor binding domain (RBD) [7,18]. The RBD is a part of the S1 subunit of the SARS-CoV-2 spike protein. While the other three vaccines generate antibodies directed at the complete spike protein that contains both S1 and S2 subunits, Walvax’s vaccine generates antibodies against RBD only (Table 1). The LNP platform of Walvax’s vaccine seems identical to Moderna and Pfizer (Table 1). Walvax’s COVID-19 vaccine can be stored for longer periods at room temperature compared to other mRNA vaccines (7 days vs. 2–24 h). The reasons for better thermostability of Walvax’s vaccine are not known.

Ultra-cold storage requirements and short shelf life slow down the distribution of Moderna and BioNTech/Pfizer COVID-19 vaccines mostly in resource poor countries of the world. Maintaining ultra-cold storage conditions is expensive and difficult to arrange in areas of the world with limited resources. Due to the short shelf life of these vaccines and vaccine hesitancy among the U.S. population, many expired COVID-19 vaccines end up discarded. Globally, about half of the vaccines are wasted due to improper temperature control [19]. One of the challenges is to develop a clinically effective thermostable mRNA vaccine, which can be stored for a longer period without high storage costs. In this review, we highlight the findings in the current literature that explore potential strategies of developing thermostable RNA vaccine delivery systems.

## 2. Stability of mRNA

An mRNA-based vaccine is a new and promising platform that can be prepared, comparatively, in a short period of time. However, its use is limited due to stability issues. mRNA is highly susceptible to RNase enzymes, which degrade it easily [11]. Therefore, it requires extreme sterile conditions of an RNase-free environment in preparation, storage, and administration. All equipment used for these three stages must be sterile. However, contrary to popular belief, RNA is considerably thermodynamically stable in vitro [11]. RNA is more thermostable, but more susceptible to oxidation than DNA [20]. It is the only biological molecule that can be heated (up to 90 °C) or frozen, lyophilized, precipitated, or resuspended in aqueous media without damage. Lyophilized RNA becomes active upon resuspension with water. In vitro stability of RNA is reduced by RNase enzyme in the vehicle [11]. RNA is susceptible to hydrolysis at a pH higher than 6. RNA solutions prepared RNase-free, neutral to a slightly acidic medium, have good stability [11]. In vivo, RNA is degraded by RNases, endonucleases, and 5′ exonucleases [21]. There are also several cofactors, such as helicases, polymerases, and chaperones, which are responsible for RNA degradation [21]. Encapsulation of mRNA by LNP is believed to provide protection against enzymatic degradation. However, RNA hydrolysis may increase in LNP if cationic lipid lowers the pKa of ribose 2′ hydroxyl group [22]. For RNA vaccines, it is critical to protect the integrity of the complete molecule as a single abnormality can stop translation [23].

## 3. Strategies for Enhancing mRNA Vaccines Stability

Optimization of an mRNA sequence and vector selection play important roles in the development of a stable mRNA vaccine formulation. The removal of the aqueous phase by freeze-drying can also play a critical role in the stability. In the following section, we briefly discuss three approaches that have been successfully employed to stabilize mRNA vaccines. 

### 3.1. mRNA Modification

Mature mRNA contains a protein-encoding region (open reading frame) and non-coding regulatory regions, including 5′ cap structure, 5′ and 3′ untranslated regions (UTRs), and poly(A) tail at the 3′ terminus. For vaccine preparations, mRNA is transcribed in vitro and the sequence of the synthetic mRNA is optimized as part of formulation development. Both coding and non-coding regions can be optimized or modified to improve the stability, immunogenicity, and translation efficiency of mRNAs [9]. Protein-encoding regions can be modified by different approaches, including incorporation of nucleoside-modified RNA, sequence-engineered mRNA, and self-amplifying mRNA. For example, replacement of uridine with pseudouridine in the coding region reduced degradation by RNase in a nucleoside-modified approach [9]. Both Moderna and Pfizer modified the mRNA encoding viral spike protein by replacing natural residues with two consecutive prolines at amino acid positions K986 and V987 for their COVID-19 vaccines [5,6,8]. The modification allowed stabilization of the spike on the virus particle in its ‘pre-fusion’ conformation. An example of sequence-engineered mRNA is maximizing guanosine/cytosine content in the protein-encoding region of mRNA. Self-amplifying mRNA encodes the desired antigen and viral replicase, which significantly increase antigen expression. 

Modifications of the non-coding regulatory regions also increase resistance against degradation by RNase and exonuclease. Common approaches, such as 5′-cap modification and elongation of poly(A) tail, have been applied to optimize COVID-19 mRNA vaccines [24]. The processes of mRNA modification have been discussed elsewhere in more detail [9,14,25]. Although all of these modifications can potentially improve the in vivo stability of mRNA, naked mRNA without any carrier has limited use in vaccination due to poor cell penetration.

It should be noted that the COVID-19 vaccine (CVnCoV) developed by CureVac used sequence optimized, capped, chemically unmodified mRNA [16]. It is not clear whether the use of unmodified mRNA helped improve thermal stability of CureVac’s vaccine. Initial studies from CureVac revealed frequency and intensity of adverse effects—headaches, fatigue, myalgia, and chills increased with higher dose levels, which prompted the use of a low dose (12 µg) for the clinical trial [17]. Pfizer initially tested both unmodified (3–30 µg) and modified mRNA (10–100 µg) vaccines in clinical trials and found that unmodified mRNA produced adverse effects, such as local site reaction, fever, and fatigue at lower doses than the modified ones [6]. Although likely more thermostable, the adverse effects of vaccines based on unmodified mRNA are major shortcomings. Based on the low efficacy of their first-generation COVID-19 vaccine [17], CureVac later designed a second-generation vaccine (CV2CoV) containing unmodified mRNA with multiple changes in the noncoding region. The CV2CoV showed increased protein expression in vitro and robust immunogenicity in preclinical studies compared to CVnCoV [26,27].

### 3.2. Viral and Non-Viral Vectors

Naked unprotected mRNA has poor cellular absorption due to its hydrophilic nature, high molecular weight, and high negative charge. Therefore, encapsulation of mRNA in different carriers or vectors has been studied to increase intracellular delivery. Vectors can be viral or non-viral. Viral vectors are genetically modified viruses containing genes of a virus partially or completely replaced by antigen-encoding RNA. Different RNA viruses, such as alphavirus, picornavirus, and flavivirus, have been used as vectors for mRNA delivery [28,29,30]. Some disadvantages of viral vectors include possible allergic reactions and host genome integration of viral vectors [31,32].

Non-viral vectors can be classified as polymer-based, lipid-based, or hybrid. Biodegradable polymers are commonly used to develop nanoparticle formulations. However, biodegradable polymers such as poly(lactic-co-glycolic acid) (PLGA), cannot effectively encapsulate negatively charged mRNAs [25]. To form a stable complex with mRNA, positively-charged cationic polymers, such as polyethylenimine (PEI) or its derivatives, are more effective [33]. These hydrophilic polymers have been used as mRNA vectors, but their use has been limited due to toxicity. The toxicity was reduced by conjugating PEI with cyclodextrin [34]. Polymer-based mRNA delivery is also limited by poor correlation between polymer structure and transfection ability.

LNPs are more commonly used for RNA delivery [6,11,14,16]. Pfizer and CureVac’s COVID-19 vaccines use a LNP platform developed by Acuitas Therapeutics [11], which contains ionizable lipid, PEG lipid, cholesterol, and neutral/helper lipid (Table 1). Ionizable lipids are cationic at acidic pH, but they have a neutral charge at the physiological pH. This property is utilized to encapsulate mRNA at acidic pH and to reduce toxicity at neutral pH in the physiological environment. The ionizable lipids are composed of an amine head group and multiple hydrophobic tails joined together by a linker group. Ionizable cationic lipids have been widely studied for delivery of nucleic acids [25]. The mRNA COVID-19 vaccine developed by Moderna used a proprietary ionizable lipid (SM-102) to form a stable complex (lipoplex) with the negatively charged mRNA. Ionizable lipid developed by Moderna has increased biodegradability and drives immunogenicity of their LNPs [35]. Faster elimination of biodegradable ionizable lipid improves tolerability without compromising immune response [36].

PEG lipids are added to prevent aggregation, thereby improving physical stability of LNP. Other helper lipids, such as cholesterol and 1,2-distearoyl-sn-glycerol-3-phosphatidylcholine (DSPC), are added to the formulation to improve rigidity of particles [9,37], therefore improving integrity in vivo (Table 1). Among the different kinds of vectors, the stability of the mRNA LNP complex is poorly understood. The LNP can be unstable during freeze–thaw cycles and their shapes and encapsulation capacity may change at high temperature [11]. Components of LNP, such as DSPC and ionizable lipids, are susceptible to temperature and pH-dependent hydrolysis [14]. Impurities in PEG lipid may cause oxidation of cholesterol used in the LNP [38].

Storage guidelines from EMA indicate that both Moderna and BioNTech mRNA vaccines are stable in frozen condition up to 6 months at −25 °C, up to 30 days at refrigerator temperature (4 °C), and up to 6 h at room temperature. A COVID-19 mRNA vaccine candidate from Walvax (ARCoV) claims to retain in vivo delivery efficiency for at least 7 days when stored at room temperature [7]. However, the transfection efficiency was significantly decreased after storing at 37 °C for 7 days. Besides vaccines, LNP containing ionizable lipid, cholesterol, PEG lipid, and DSPC have been utilized to deliver siRNA in FDA approved formulation of Onpattro [14]. The shelf life of Onpattro is 3 years at refrigerator temperature (2–8 °C), indicating the extreme cold storage temperature for COVID-19 mRNA vaccines is dictated by the unstable nature of mRNA, not the instability of LNP.

Apart from ionizable lipids, cationic lipids composed of tertiary or quaternary amines have been used as carriers of mRNAs. In one study, nanostructured lipid carriers consisting of cationic lipid (DOTAP) alone showed stability of particle size and lipid composition for two years at refrigerator temperature. Upon complexation with RNA, the nanostructured lipid carrier was able to maintain RNA integrity of the complex only for five weeks at refrigerator temperature [39]. Cationic lipids tend to have reduced efficacy due to possible protein binding in vivo. Furthermore, inflammatory reaction and toxicity are observed with cationic lipid formulations [40,41].

In LNP, mRNA forms a complex with ionized lipids, whereas liposomes have a lipid bilayer with an aqueous core. Degradation of mRNA is higher in the presence of water, which makes liposomes less likely to be stable than LNP [14]. Notably, a stable mRNA vaccine was developed with liposomes [42], but it is not known whether the shelf life of liposomal vaccines will be longer than LNP vaccines. 

Lipid-polymer hybrid nanoparticles (LPHNP) represent a potential delivery system that combines the merits of both systems [25]. The LPHNP has shown promising initial results in delivering mRNA [43,44,45]. Common lipids used in LPHNP are 1,2-Dioleoyl-3-trimethylammonium propane, 1,2-dilauroyl-sn-glycero-3-phosphocholine,1,2-distearoyl-sn-glycero-3-phosphocholine, lecithin, DSPE, and PEG-lipids. On the other hand, PLGA, polycaprolactone, polylactic acid are commonly used polymers in LPHNP. Additionally, a peptide-based vector (RALA) has been combined with a polymer (PLA) to develop peptide-polymer hybrid nanoparticles for mRNA delivery [46]. The stability of an mRNA vaccine developed by these promising hybrid systems is still unknown.

Cationic peptides, such as protamine, have also been used to develop early mRNA vaccines. A rabies vaccine developed using a protamine-mRNA complex showed outstanding thermostability [47]. The lyophilized vaccine was stable for 12 months (Table 2) when stored at −80, 5, or 25 °C without loss of efficacy in mice. The mRNA and protamine complex was based on CureVac’s proprietary RNActive technology. The lyophilized vaccine was even stable up to 3 months at 70 °C. This lyophilized formulation showed much higher thermostability at room temperature than LNP-based vaccine formulations (Table 2). Unfortunately, the protamine-based rabies vaccine required specialized injection devices in a human clinical trial to produce protective titer. In contrast, the LNP-based rabies vaccine showed better clinical efficacy in both animal and human trials [16], suggesting that thermostability and clinical efficacy do not always complement each other. 

### 3.3. Freeze-Drying

Freeze-drying or lyophilization is a commonly used technique to improve stability of liquid vaccine formulations [50,51]. As the name implies, the vaccine is first frozen and the aqueous solvent is removed through a sublimation and desorption process. It should be noted that lyophilization is a complex, expensive, and slow process that involves additional steps, such as drying and reconstitution. Jones et al. reported that freeze-dried replicating RNA can be stable for 10 months at 4 °C when trehalose is added as cryoprotectant [48]. 

In laboratory settings, freeze-drying has been utilized to improve the shelf life of LNP mRNA vaccine formulations (Table 2). Lyophilized LNP containing RNA retained biophysical properties and in vivo protein expression ability for 21 months at 4 °C and for 8 months at room temperature [39]. Due to lack of published data on thermostability of LNP formulations with mRNA, we looked at stability data of LNP formulations that were prepared to deliver small interfering RNA (siRNA) [12,52]. The siRNA is primarily utilized in pharmacotherapy, but not considered for prophylactic vaccines due to its gene silencing nature [53]. LNP complexed with SiRNA was stable for about 5 months at 2 °C, where stability of encapsulated siRNA was indicated by in vitro studies [12]. Upon lyophilization, LNP maintained in vitro gene silencing efficacy of siRNA for 11 months at −80 °C. Aggregation and loss of efficacy were observed by freeze–thaw cycles in these formulations, which were prevented by the addition of lyoprotectants, such as sucrose and trehalose. The lyoprotectants also facilitated reconstitution of lyophilized formulation with water instead of ethanol [12]. 

Interestingly, loss of in vivo mRNA delivery efficiency was reported following lyophilization of LNP in one study [49]. It is apparent that optimization of freeze-drying conditions, selection of the lipid vector, and concentration of lyoprotectants are important considerations to achieve a stable product. It should be noted that both Moderna and BioNTech/Pfizer vaccines use sucrose as a lyoprotectant in their frozen vaccines. Therefore, lyophilization could be the next logical step towards improving stability. 

Currently, Moderna and Pfizer COVID-19 vaccines are supplied as frozen and stored at −80 °C. Current literature suggests that lyophilization may allow these vaccines to be stored at refrigerator temperature (2–8 °C) instead of subzero temperature. It was reported that Pfizer is considering lyophilization for their COVID-19 vaccine [54]. However, it is not known if the lyophilization process itself will impact the potency and efficacy of the COVID-19 mRNA vaccines. Alternative drying processes, such as spray drying, foam drying, spray-freeze drying, vacuum drying, and supercritical fluid drying have been successfully employed to improve stability of biopharmaceutical formulations, including vaccines [55,56], which could also have the potential to improve stability of mRNA vaccines. 

## 4. Conclusions

The field of mRNA vaccine development is rapidly evolving. At present, many mRNA vaccines are in clinical trials for different diseases, including influenza and cancer [9]. One of the major challenges of mRNA vaccines is the lack of thermostability and ultra-cold storage requirement. A thermostable vaccine (CvnCoV) from CureVac showed clinical efficacy below 50% [17], which indicates that developing a thermostable vaccine with ‘high clinical efficacy’ remains a challenge for the time being [17]. In this review, we have outlined the stability and storage data of various mRNA COVID-19 vaccines. The stability data of mRNA vaccines are rare in the literature, indicating well-designed mechanistic studies using different excipients, storage temperatures, and longer periods of time should be conducted to fill the existing gaps in knowledge. Based on the limited information available, we reached a conclusion that the lack of thermostability is related to the fragile nature of synthetic mRNA used in these vaccines.

We also concluded that the thermostability of mRNA vaccines can be improved, with or without major changes in the formulation. For example, lyophilization or alternative methods of drying of mRNA LPN suspension will be viable options to improve the stability and storage conditions of mRNA vaccines. Secondly, thermostability of vaccines can be improved by optimizing the mRNA sequence. After initial failure in the clinical trial with their first thermostable vaccine, CureVac is now pursuing a second COVID-19 vaccine with mRNA optimized in the noncoding region [26]. Thirdly, the excipients, such as lipid and cholesterol in LPN, can make the vaccine prone to oxidative degradation; thus, stability of vaccines can be improved by optimizing the components and manufacturing process of nanoparticles. Finally, a theoretical study suggests that redesigning mRNA to form double-stranded regions could be another option to improve stability of vaccines [22]. Overall, many potential strategies remain to be explored to overcome the challenges of stability and storage of COVID-19 mRNA vaccines.

## Figures and Tables

**Table 1 vaccines-09-01033-t001:** Comparison between mRNA vaccines against COVID-19.

Developer	Moderna	Pfizer/BioNTech	CureVac	Walvax
Name	mRNA-1273	BNT162b2	CVnCoV	ARCoV
Country of origin	USA	USA and Germany	Germany	China
Active ingredient	Modified mRNA encoding viral spike glycoprotein	Modified mRNA encoding viral spike glycoprotein	Unmodified natural mRNA encoding viral spike glycoprotein	mRNA encoding viral receptor binding domain of spike glycoprotein
Vector	LNP	LNP	LNP	LNP
Excipients	Ionizable cationic lipid (SM-102)	Ionizable cationic lipid (ALC-0315)	Ionizable lipid (unknown)	Ionizable lipid (unknown)
	Helper lipids (DSPC, Cholesterol)	Helper lipids (DSPC, Cholesterol)	Helper lipids (phospholipid, cholesterol)	Helper lipids (DSPC, Cholesterol)
	PEG lipid (PEG-DMG)	PEG lipid (ALC-0159)	PEG lipid	PEG lipid (PEG-DMG)
	Buffer (Tris)	Buffer (monobasic potassium phosphate, dibasic sodium phosphate dihydrate)	No data	Citrate buffer
	Salt (Sodium acetate)	Salt (Sodium chloride, potassium chloride)	No data	No data
	Sucrose	Sucrose	-	-
pH	7–8	7–8	No data	No data
Diluent	None	Mixed with saline before administration	Saline	Saline
Shelf life (Frozen state)	Up to 6 months at −20 °C	Up to 6 months at −80 to −60 °C	Up to 3 months at −60 °C	No data
Shelf life (2–8 °C)	Up to 30 days	Up to 5 days or 31 days (EMA guideline)	Up to 3 months at 5 °C	No data
Shelf life (Room temp)	Up to 12 h	Up to 2 h before dilution and up to 6 h after dilution	Up to 24 h	Up to 7 days

**Table 2 vaccines-09-01033-t002:** Stability data of RNA formulations.

Reference	RNA Delivery	Suggested Shelf Life	Evidence of Stability
[48]	Self-amplifying RNA	Lyophilized: 10 months (4 °C)	In vitro protein expression, in vitro transfection efficiency
[49]	RNA in LNP	3 months at liquid nitrogen	In vitro and in vivo mRNA expression, particle size
[47]	mRNA-protamine complex	Lyophilized: 12 months (25 °C)	Antibody analysis, challenge infection
[7]	mRNA in LNP	7 days (25 °C)	In vivo delivery efficiency
[39]	Self-amplifying RNA or mRNA in LNP	5 weeks (4 °C)	Particle size, RNA integrity by gel electrophoresis, In vivo protein expression
		Lyophilized: 8 months (room temperature), 21 months (4 °C)	

## Data Availability

Data are available in correspondent journal.

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
