# Peer review of "Challenges of Storage and Stability of mRNA-Based COVID-19 Vaccines"

_vaccines, 2021, doi:10.3390/vaccines9091033_

Round 1

Reviewer 1 Report

The submitted manuscript entitled  ‘Challenges of storage and stability of mRNA based COVID-19 2 vaccine’ reports a summary of studies /understanding on the stability of mRNA vaccines for covid. In general terms the manuscript is easy to follow, and is a topical subject.
However, in my opinion, author should provide  more information in depth  around the topic – stability: eg   information on why mRNA is not stable chemically, and how formulation helped or damaged which would be very useful for vaccine development. It is great to have the section on mRNA modification.  In addition, the review did not give a clear picture how to improve the stability. The conclusion can be more elaborated.   In a number of occasions authors need references to back up their statements. Some examples are included below.

Specific points:  

COVID-19 is an abbreviation, should be defined on first use

Line 23 – importantly, to date there is no evidence showing ‘it spread globally’ from China, and there are no references supporting their statements. Suggest to remove the words

Line 26 ‘Since November 2020, six vaccines have been introduced globally’. In addition timeframe for this sentence should be defined

Line 85-line 86 are not relevant to the subtitle.

Table 2 si-RNA are listed and reason was not justified

Lines 129-130 contains interesting information but is not clear how the shelf life defined, was it mRNA or particles.

Line 134: ‘Both lipid nanoparticles and liposomes have been used to develop mRNA vaccines; in line 134 is not needed there as in the last paragraph there is discussion about LNP. Interestingly author suddenly stopped using LNP for ‘lipid nanoparticles’. 

To conclude, I think that there is some merit in this study major revision is required.

Reviewer 2 Report

The review article by Uddin and Roni generally is very well written and provides a nice overview on the current challenges of storage and stability of mRNA based COVID-19 vaccines.

My major criticism relies on the fact that the review mainly focuses on the two most effective mRNA vaccines that are currently in use- the Moderna and Pfizer-BioNTech COVID-19 vaccines. Since these are the most effective COVID-19 vaccines on the market and the only mRNA vaccines currently approved by regulatory authorities to be used under EUA, I understand the author’s rationale to only focus on these two vaccines. However, as the authors most likely are aware of, there is a third company that developed a promising mRNA COVID-19 vaccine early on during the pandemic: The mRNA vaccine developed by CureVac. Unfortunately, first reports on a clinical trial enrolling about 40,000 people suggest a disappointing efficacy of only 47%. If this number holds true, it might not be awarded authorization by regulatory authorities. The reasons for the unexpected low efficacy are unclear and different hypotheses are currently under debate (see for instance the article by Jon Cohen in the 25 June 2021, Vol 372 Issue 6549 of Science (“What went wrong with CureVac’s mRNA vaccine”). The reason why I mention CureVac’s mRNA vaccine is mainly because it holds some practical transportation and storage advantages over the rival mRNA vaccines from Moderna and Pfizer, in that it can be stored longer at higher temperatures. This is also reviewed in the following publication: Journal of Pharmaceutical Sciences 110 (2021) 997-1001: Addressing the Cold Reality of mRNA Vaccine Stability by Daan J.A. Crommelin, TJ Anchordoquy, DB Volkin, W Jiskoot and Enrico Mastrobattista. According to information released by CureVac, the vaccine is stable at 2-8 °C for at least 3 months and claimed to be stable at room temperature for up to 24 h. (references are provided in the article mentioned above). More information on CureVacs vaccine can also be found in the article “mRNA-based SARS-CoV-2 vaccine candidate CVnCoV induces high levels of virus-neutralising antibodies and mediates protection in rodents, by Rauch et al..; NPJ Vaccines (2021) 6:57 ; https://doi.org/10.1038/s41541-021-00311-w “ It therefore appears to be important to compare in your review article CureVac’s vaccine formulation and design (as far as it is known) with that of the Moderna and Pfizer mRNA vaccines and discuss what might be the differences that render CureVac’s vaccine more stable at higher temperatures as compared to those made by the rivals. I therefore highly recommend modifying the review article and include a discussion of CureVacs mRNA vaccine platform as well. Likewise, the study by Zhang et al.(Cell, 2020, 182, 1271-1283) mentioned in your reference 4, is not further discussed, even though the COVID-19 mRNA vaccine appears to be stable for up to 7 days at RT. Did the authors of that study discuss what might confer the good thermostability?. I recommend to also include this study in your discussion and compare the formulation and design of the mRNA vaccine from China with the others.

Minor comments:

Line 24: change “appear” to “appears” or “appeared”.

Table 1: For easier comparison, I suggest to have the same excipients being listed in the same row in both columns (for instance: Ionizable Cationic lipid should be side by side in both columns. Instead, cationic lipid in the Moderna column is shown in the same row as Helper Lipids in the Pfizer column. There seems to be a gap between Lipid nanoparticle and Ionizable cationic lipid in the Moderna column that caused this mismatch. Please fix.

Line 77: enzymatic degradation: Please specify which enzymes likely cause the degradation.

Line 153 besides the study by Jones references in your manuscript (10), you might also want to discuss the study in which a mRNA-protamine vaccine candidate against rabies was freeze-dried and tested in mice. The study analyzed different storage conditions (Stitz L, Vogel A, Schnee M, et al. A thermostable messenger RNA based vaccine against rabies. PLoS Negl Trop Dis. 2017;11(12):e0006108.)

Round 2

Reviewer 1 Report

The manuscript has been significantly improved. But still there are many issues required to be addressed:

Specifically:

Abstract: Do you mean  ‘a  few vaccines’ instead of   ‘few vaccines’

Table 1: “Mixed with 0.9% normal saline before administration” can be replaced with “Saline ‘ ; ‘Up to 2 hours and up to 6 hours after dilution; is not clear;  ‘Ionizable lipid ‘ may be changed so to reflect it is unknown   

Line 55 or 61: it is better to indicate the development phase for this vaccine. As in Table 1  for both cases ‘ Ionisable lipid’ is not defined, it is hard to be convinced that the formulations of the two vaccines are same. If they are really same, correct Table 1.  Regarding efficacy CureVac’s COVID-19 vaccine  dose is much smaller than Pfizer’s, it should be included.    

Line 134 ‘adverse effects;’ should be specified. Same for line 137

Line 199: still unclear whether the RNA stable or not. And what change to the RNA complex. It should be described like Table 2 – evidence of stability

Line 219: the long stability should not be attributed to only those factors – as Self-amplifying RNA is generally more stable

Table 2: siRNA are included. Why?

Line 271 -272 is not clear

The conclusion section needs condensation. The last paragraph seems irrelevant to the topic anymore   

Reviewer 2 Report

The authors made the modifications I recommended, and the quality of the review has improved significantly.

Minor comments:

Line 63: “ …undergoing phase III clinical trial “  please change to “…… undergoing a phase III clinical trial”

Line 65: The authors are correct in pointing out that the mRNA vaccine developed by Walvax encodes the viral receptor binding domain (RBD). While it is true that it is different from the mRNA from the other three companies mentioned in table 1, the target is still the same in that the RBD is part of the  SARS-CoV-2 S protein.  Thus, all vaccines will generate antibodies directed at the RBD.  The difference is that the other three vaccines also elicit antibodies directed epitopes outside of the RBD. Please modify the text to avoid a misunderstanding.

Line 68:  7 days vs 2-12 hours: according to table 1, the other mRNA vaccines can be stable at RT for 2-24 hours. Please adjust accordingly

Line 84: “.. susceptible to RNAse enzyme which degrade it easily” please change to “  .. susceptible to RNase enzymes which degrade it easily”

Line 87/88: Do you mean: “ RNA is more thermostable but more susceptible to oxidation than DNA” ?

Line 94: RNases

Line 116: …. Pseudouridine in coding region…. Please change to:  ..Pseudouridine in the  coding region….

 Line 134:  please change to : … at a higher dose of vaccine…

Line 135: please change to : .. prompted the use of a low dose (12 ug) for the clinical trial…

Line 163: change “used LNP platform”   to: “use an LNP platform”  

Line 184: change “ indicates” to: “indicate”

Line 203:  please change  “mRNA form” to “mRNA forms”

Line 217:  please change “Cationic peptide like protamine has also been…. “ to : “Cationic peptides like protamine have also been…. “
